# SHADOWPUNCH: FAST ACTIONS SPOTTING BENCHMARK

## ABSTRACT

We introduce an open dataset for video event spotting focused on fast-paced events in shadowboxing videos captured at high frame rates. The dataset features accurate frame-level annotations for diverse punch types alongside pose keypoint annotations, enabling the development of robust event recognition models. This work presents a novel benchmark exploring two distinct approaches to event spotting: direct prediction from image data and a staged approach involving intermediate pose estimation followed by event detection based on the detected keypoints. We provide baseline neural network solutions incorporating temporal information for both tracks, facilitating comparative analysis of these methodologies. This shadowboxing dataset advances the field of automatic sports analysis and contributes to the broader understanding of video events recognition.

## 1 INTRODUCTION

Understanding and interpreting video content is a fundamental challenge in computer vision with vast real-world implications. From autonomous driving and surveillance systems to sports analytics and human-computer interaction, accurate video analysis enables us to extract meaningful information about events and actions occurring within visual data streams. One critical task of video understanding is event spotting: the task of precisely identifying specific event of interest occurrences within a video sequence. Specifically, this involves highlighting the exact frame in which a fast-paced event occurs.

However, accurately spotting fast-paced events remains a significant challenge because it often requires detecting minute features within vast amounts of data from two modalities: spatial and temporal. Existing approaches often struggle with the high temporal resolution required to capture subtle movements and nuances characteristic of rapid actions. Moreover, many datasets used for benchmarking event detection either focus on slower-paced activities or lack detailed frame-level annotations essential for training robust models.

To address these critical needs and challenges, we introduce ShadowPunch, a novel open dataset specifically designed for video event spotting in the sports domain of shadow boxing. Shadow boxing is a high-intensity activity characterized by rapid punches and intricate movements, making it an ideal testbed for developing robust event recognition models.

Our dataset features accurate frame-level annotations for a diverse range of punch types, including jabs, hooks and uppercuts. Furthermore, ShadowPunch includes pose keypoint annotations, capturing the positions of key body joints in different moments of the video sequences. This additional layer of information allows researchers to explore sophisticated event recognition pipelines that leverage intermediate spatial cues. By providing this rich and comprehensive dataset, we aim to empower the computer vision community with a valuable resource for developing and evaluating advanced models capable of accurately spotting fast-paced events in video data.

This work makes the following key contributions to the field of event recognition:

- **Introduction of ShadowPunch dataset**: We present ShadowPunch, a novel open dataset for video event spotting in the domain of shadowboxing. This high-intensity activity presents unique challenges due to rapid movements. ShadowPunch provides accurate frame-level annotations for various punches and incorporates pose keypoint annotations.

- **Benchmark for Comparing Event Spotting Approaches**: We propose a novel benchmark comparing two distinct methodologies for event spotting: direct prediction from image data and a staged approach incorporating intermediate pose estimation. This benchmark allows researchers to directly compare the effectiveness of end-to-end versus staged approaches for event spotting.
- **Baseline Solutions**: To facilitate further research, we provide baseline implementations for both the direct prediction and staged approaches, providing researchers with a foundation to develop and evaluate their own methods.

The code and the dataset are publicly available at LINK.

## 2 RELATED WORK

**Video understanding tasks.** Advances in deep learning and available computing power have driven remarkable progress in video understanding in recent years. The video understanding sphere includes several related tasks ordered by their temporal resolution (coarsest to finest): Action classification involves assigning a label from a predefined list to an entire video clip based on its content. Temporal Action Segmentation (TAS) involves identifying the start and end points in time of an action within an untrimmed video sequence and assigning a class label to the action. Action spotting is a task of detecting the presence and precise temporal location of an event within a video, typically pinpointing it to a single frame and providing corresponding class label. These events are usually brief.

Hong et al. (2022) further differentiates these tasks by introducing precise temporal event spotting. This defines correct prediction as follows: The predicted event frame timestamp must fall within $\delta$ frames (a small window of just a few video frames) of the labeled ground-truth event and also have the correct class label. We will further focus on this task formulation.

**Public datasets.** Several public datasets are available for action detection or spotting in general and particularly within the sports domain; these datasets usually come with extra annotations to provide more context or in-depth information on actions. Historically, many sport-related datasets have focused on temporal segmentation of prolonged events. For example, FineDiving (Xu et al., 2022) features 3000 video samples from various diving competitions. Each annotation includes not only the type and difficulty level of the diving action but also the judges' scores.

Some datasets provide action annotations with both class and temporal boundaries along with spatial boundaries of the actor performing an action on a playground, allowing for both action analysis and multi-object tracking (MoT). For example, FineSports (Xu et al., 2024) consists of 16,000 annotated events in basketball videos, accompanied by approximately 123,000 spatial bounding boxes of players.

There are also datasets focusing primarily on event spotting tasks. For instance, SoccerNet-v2 contains a large collection of football broadcast videos (764 hours) with 110,458 annotated action timestamps over 17 in-game event types. It provides other annotations as well, which enable comprehensive football analysis, but contains no human pose data. However, it is worth noting that the videos are recorded at a resolution of 1280×720 pixels and a frame rate of 25 frames per second, making it challenging to pinpoint the exact frame of occurrence for some fast events in difficult cases such as very short ball touches.

On the other end of the spectrum, there is a high-frame-rate video dataset OpenTTGames (Voeikov et al., 2020), which features table tennis videos sampled at 1080×1920 pixels and 120 frames per second. This dataset offers annotations for 4,271 fast-paced events of three classes and also includes semantic segmentation maps of the players.

There are multiple publicly available pose estimation datasets in the sports domain, such as Human3.6M (Ionescu et al., 2014) and SportsPose (Ingwersen et al., 2023), which do not provide precise action timestamps for event spotting tasks.

However, there is a very limited set of datasets that feature both action labels and pose data. A notable recent example is FS-Jump3D (Tanaka et al., 2024), which includes 86 3D pose keypoints obtained through optical markerless motion capture and temporal action segmentation labels for figure skating jumps. Notably, all these datasets mainly contain video data collected within laboratory

environments, which limits the diversity of backgrounds and lighting conditions. The FS-Jump3D dataset, in particular, contains data from only four subjects in a single ice rink laboratory environment.

**Video understanding models.** Early approaches were based on 2D CNNs to extract spatial features from each frame independently. These frame-level features are then aggregated using temporal pooling or recurrent networks such as Long Short-Term Memory (LSTM) or Gated Recurrent Units (GRUs) to capture temporal dependencies across frames (Donahue et al., 2015). To directly model the spatiotemporal dynamics of a video, 3D CNNs extend traditional 2D convolution operations by adding a temporal dimension to the convolutional kernels. Architectures such as C3D (Tran et al., 2015) and I3D (Carreira & Zisserman, 2017) learn to represent both spatial and temporal patterns jointly, making them well-suited for action recognition and temporal action localisation.

Another approach employed a two-stream architecture introduced by Simonyan & Zisserman (2014). This involves a separate network to process RGB frames, capturing spatial appearance cues, while another stream processes optical flow inputs to focus on temporal features. Enhancements to this approach, introduced by Wang et al. (2016), integrated short-term and long-term temporal dependencies to improve robustness against variations in video content, as well as utilised extra modalities such as RGB difference and warped optical flow fields. The latter was meant to compensate for camera motion so that the optical flow field highlights features of human action rather than background.

More recent approaches adopt transformer-based architectures to model both short- and long-term temporal relations within a video. For example, TallFormer (Cheng & Bertasius, 2022) utilises a transformer-based encoder to extract short-term features from a video clip. ActionFormer (Zhang et al., 2022) uses a convolutional-based video frames encoder with the embedding features being further encoded into a feature pyramid using a multi-scale Transformer, while employing a convolutional decoder to solve the action localisation task. The ASTRA architecture combines several temporal embeddings for visual and audio modalities using a transformer encoder-decoder architecture (Xarles et al., 2023) to address the SoccerNet-v2 action spotting challenge (Deliege et al., 2021).

Meanwhile, there has been growing popularity in approaches that are not directly based on video frames but rather on features from intermediate representations such as pose skeletal data to extract temporal information from videos (Yeung et al., 2024; Ibh et al., 2024; Deyzel & Theart, 2023). For example, TemPose (Ibh et al., 2023) uses skeleton-based temporal self-attention for action prediction by utilising a transformer-based model to process pose-estimation skeleton sequences as well as encoded auxiliary data on players' positions on the court and the shuttlecock position for action recognition in badminton video. This approach minimises reliance on non-human-related visual context by decreasing the effect of background actions, as the model consumes manually crafted intermediate representations that are focused on action-specific features. These models often offer lighter computational loads due to reduced amounts of raw data to be processed.

Despite significant advances in event spotting within video understanding tasks, there remains a notable lack of fast-paced action spotting datasets, particularly for shadowboxing. Existing datasets often lack precise action timestamp annotations and human pose data. Additionally, current datasets are predominantly laboratory-based, limiting their applicability to real-world scenarios with varying lighting conditions and background complexity. Our work aims to address these limitations by introducing a new dataset with detailed shadow boxing action labels and skeletal pose information. We also propose a benchmark for evaluating event spotting models, suggesting comparison of direct prediction versus intermediate representation approaches.

## 3 SHADOWPUNCH DATASET

We introduce a comprehensive boxing dataset, ShadowPunch, designed for advanced pose estimation and action classification in boxing. This dataset comprises over 27 high-definition videos capturing a diverse array of boxing movements and techniques across 230,502 frames, recorded at 60 frames per second. This framerate, which is higher than the conventionally used 25-30 fps, provides more detailed temporal information for analysing the dynamics of fast boxing actions. An example of frame sequences with corresponding annotation is given in Fig. 1. Each video includes

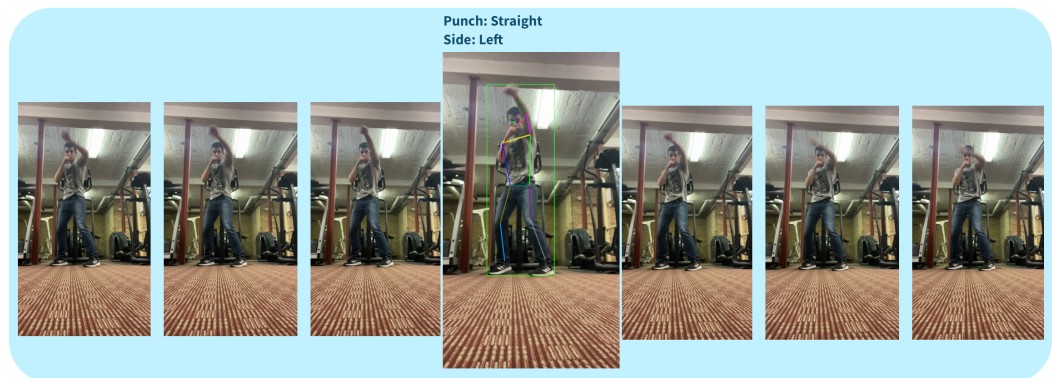

Figure 1: A sequence of frames from a dataset video with a highlighted frame annotated as a punch event, along with corresponding skeletal pose data.

recordings of multiple punches and punch series performed by boxers of diverse levels ranging from beginners to seasoned world-level professionals. This variety ensures comprehensive coverage of boxing techniques and skills, providing a rich and varied portrayal of the sport's technical aspects and broadening its applicability for research in sports science.

Each video has just one person in the field of view. The videos in the dataset were captured in different environments: indoors and outdoors, in professional training venues and residential houses (Fig. 2, more examples are provided in Appendix B). The annotation was performed by four experts with deep knowledge of the sport. Quality of the annotation was ensured by double cross-check of the labels by another annotator.

Our dataset features a combination of annotations: it includes punch actions of three types (straight, hook, uppercut), no-punch event and punch side (left or right hand). Each punch event is pinpointed to the exact video frame number in which the punch takes place and is defined as the precise moment of maximum hand extension during a punch hand movement.

Table 1: ShadowPunch dataset statistics, showing the number of frames with pose and event spotting annotation. Events column indicates the total number of events, frames - number of video frames.

| Type | Poses | Events | Left | Right | Hook | Uppercut | Straight | No punch | Frames |
|------|-------|--------|------|-------|------|----------|----------|----------|--------|
| Train | 3584 | 3744 | 1557 | 1395 | 678 | 1397 | 878 | 790 | 174754 |
| Test | 1177 | 1096 | 423 | 341 | 321 | 225 | 218 | 332 | 55748 |
| Total | 4761 | 4840 | 1980 | 1736 | 999 | 1622 | 1096 | 1122 | 230502 |

The dataset features 4761 frames with annotated poses with a 17-keypoint skeleton model, and the number of annotated poses is roughly equal to the number of annotated actions. Notably, about half of these annotations coincide in the same frames, providing a unique opportunity to analyse specific postures adopted during various punches by different athletes. This overlap between actions and poses within the same frames can be beneficial for understanding movement execution and offers invaluable data for in-depth biomechanical studies. Detailed statistics on the dataset are provided in Table 1. The high-speed nature of boxing punches is evident from the dataset, which has more than one event per second on average. The data are spread over 21 videos in the Train subset and 6 videos in the Test subset.

In addition, video frames with pose estimation annotations are also equipped with the boxer bounding boxes. These bounding boxes can be used for cropping regions of interest before further processing with a deep learning model.

One challenging aspect to boxing shadow fight movements is that some actions may appear similar to the beginning of punches but are actually dives or other sport-specific movements, which can confuse some computer vision models leading to false-positive predictions. To facilitate addressing

Figure 2: Several screenshots from different videos in the dataset, showcasing the diversity of background settings and individuals performing shadowboxing.

this issue, our dataset also includes annotations for no-punch events, marking moments when an actual punch should not be predicted.

In summary, our comprehensive boxing dataset provides high-resolution temporal and spatial information, extensive expert annotations, and a diverse participant pool, making it an invaluable resource for research in human motion dynamics and the specificities of boxing techniques. Potential applications include:

- Pose Estimation Studies: Evaluate pose estimation algorithms using detailed skeletal annotations for humans performing rapid movements.
- Action Spotting Models: Research machine learning models to accurately classify and temporally spot boxing movements.
- Sports Science Research: Analyse the biomechanics of different punch types to improve training methods.

# 4 BENCHMARK

We propose the dataset and benchmark for comparison between two families of approaches: a direct approach that predicts punch events directly from the imagery data, utilising its temporal modality; or, alternatively, a multistage approach with an intermediate representation of skeletal keypoints, where final action spotting predictions are based on a sequential stream of pose keypoints.

By directly predicting punch events from raw imagery data, the model can learn a comprehensive representation of temporal dynamics without relying on artificially predefined forms of intermediate representations. This approach may be more robust to noise in the input data since it does not depend on the accuracy of an intermediate step like pose keypoint detection. Additionally, such models may have lower inference latency and could potentially be more efficient for real-time applications. However, the direct approach may be more difficult to extend with auxiliary tasks and may lack interpretability because they do not provide an intermediate representation that could offer insights into the reasoning behind the predictions.

On the other hand, the multistage approach allows for greater modularity, where each stage can be optimised independently, potentially improving the pipeline prediction quality and offering some insights via the intermediate representation. This can be particularly useful in scenarios where detailed analysis of individual stages is necessary. For instance, evaluating the accuracy of keypoint detection separately from action spotting could help identify specific areas that need improvement. Nevertheless, cumulative error propagation can occur, with errors from earlier stages (e.g., keypoint detection) propagating through subsequent stages and amplifying, leading to poor final predictions.

Given the arguments for and against both options, it seems to us a non-trivial to predict without further experimentation which approach will be optimal. Although the answer in actual application may depend on various specific aspects of particular implementations and business requirements, the general question of whether a direct approach or a staged approach with artificially defined intermediate representation may achieve better results seems an interesting research direction. Therefore,

we open this benchmark for the wider computer vision community to facilitate further investigation and advance the state-of-the-art in action spotting.

In this work, we evaluate the performance of our punch event spotting models using several well-established metrics: $F_1$-score, confusion matrix, accuracy, and frame displacement. The $F_1$-score is the harmonic mean of precision and recall, providing a balanced measure between false positives and false negatives. It is computed as:

$$F_1 = 2 \cdot \frac{Precision \cdot Recall}{Precision + Recall}$$

where precision is defined as $Precision = \frac{TP}{TP+FP}$ and recall as $Recall = \frac{TP}{TP+FN}$, where TP - total count of true positives, FP - false positives count, FN - false negatives count (Powers, 2011).

Accuracy is another metric we consider. It is defined as the ratio of correct predictions to total predictions:

$$Accuracy = \frac{TP + TN}{TP + TN + FP + FN}$$

Although accuracy provides an overall measure of the model's performance, it can be misleading when dealing with imbalanced datasets (Powers, 2011). We also use a confusion matrix, which summarizes the model's performance classification capability Bishop (2006).

Depending on the specific calculation, identifying the exact frame where the punch took place is crucial for accurate event detection. One approach to measure temporal precision is through frame displacement, which is defined as the difference between the predicted frame and the actual ground truth frame number:

$$\Delta t = |t_{predicted} - t_{groundtruth}|$$

Determining the correct frame can be particularly challenging in the presence of fast movements and sequences involving double punches of the same type. These scenarios can lead to false positives, false negatives, or significant temporal deviations between punch events. Depending on the specific model architecture, heuristics, and hyperparameters must be carefully identified to accurately pinpoint the best frame for detecting the punch.

## 5 BASELINE

The baseline architectures are not optimised to reach the best possible metrics; their main purpose is to provide the community with a reasonable quality simple starting point solution, while leaving further improvements in prediction accuracy in both tracks to the community.

### 5.1 DIRECT METHOD

The end-to-end baseline architecture employs a ResNet34-3D backbone (Tran et al., 2018), which has been pretrained on the Kinetics-400 dataset (Zisserman et al., 2017). The input to the model comprises rescaled stacks of seven consecutive video frames at a resolution of 224x224 pixels, normalised to the range [0..1]. These stacks are sampled from the dataset such that the annotated events coincide with the last frame in each stack. To maintain simplicity and reproducibility of the baseline, no data augmentation techniques were applied.

The output tensor from the backbone is processed through two linear layers to generate two prediction vectors: one consisting of four elements for punch type classification and another containing two values for determining the punch side.

A Softmax layer serves as the final activation function. The model was trained using Focal loss (Lin et al., 2017) with parameters $\alpha = 1.0$ and $\gamma = 2.0$. Loss values were computed independently for both branches (punch type and side classification) and then summed with equal weights. Training

was conducted using the AdamW optimizer (Loshchilov & Hutter, 2019) over 50 epochs with a learning rate of 0.0001, selecting the best epoch checkpoint based on the highest accuracy achieved during validation.

## 5.2 STAGED APPROACH

The staged approach involves a two-step pipeline for punch type and side classification using human pose keypoint extraction followed by event classification.

In the first step, 17 keypoints (e.g., nose, shoulders, elbows, wrists) are extracted from each frame of the video using the DEKR-HRNet architecture (Geng et al., 2021), pretrained on the COCO dataset (Lin et al., 2014). The DEKR-HRNet architecture was utilised through the MMPose framework (OpenMMLab, 2020). Keypoints are detected for seven consecutive frames, sampled similarly to the direct method. Their x and y coordinates are normalised by dividing by the width and height of the video frames, respectively. After normalisation, the input tensor becomes B×34×7, where B is the batch size (16 in our case), 34 represents the 17 keypoints with their respective x and y coordinates, and 7 is the number of frames in the sequence.

In the second step, the normalised keypoints are passed into an architecture designed to capture temporal dynamics. A linear projection layer first transforms the keypoints into a higher-dimensional space (512 dimensions). The sequence is then processed by a Transformer encoder (Vaswani et al., 2017), which models temporal dependencies across the frames. Our model uses a Transformer with 16 attention heads and two layers. Then, attention pooling, inspired by the self-attentive pooling mechanism from Chen et al. (2023), is applied to weigh the most relevant frames, aggregating them into a single vector representation. The pooling layer is followed by Dropout with $p = 0.3$. This same Dropout probability is also applied in the Transformer backbone. Finally, the resulting feature map is processed by two linear layers to produce predicted probabilities for punch type and side predictions, similar to the direct method described above. Layer normalisation was applied after the linear projection and transformer layers to stabilise training and improve convergence.

The loss function, optimizer, and training strategy were the same as those used in the direct method, with the only difference being the total number of epochs. For this approach, the model was trained for 250 epochs.

## 6 RESULTS AND DISCUSSION

### 6.1 DIRECT METHOD

In the direct method, we trained a deep neural network to classify punch types and sides directly from the video frames stacks without using intermediate representation. The approach yielded an overall accuracy of 94.89%. The side classification achieved an accuracy of 99.45% (See Tab. 2 and 3). Confusion matrices are presented in Fig. 4.

Table 2: Punch Type classification metrics for the direct method.

| Class | $F_1$-Score | Precision | Recall |
|---|---|---|---|
| Hook | 0.9654 | 0.9721 | 0.9587 |
| Uppercut | 0.9091 | 0.8494 | 0.9778 |
| Straight Punch | 0.9377 | 0.9862 | 0.8938 |
| No Punch | 0.9774 | 0.9789 | 0.9759 |

Table 3: Side Classification Metrics for the direct method.

| Class | $F_1$-Score | Precision | Recall |
|---|---|---|---|
| Left | 0.9965 | 0.9930 | 1.0000 |
| Right | 0.9956 | 1.0000 | 0.9912 |

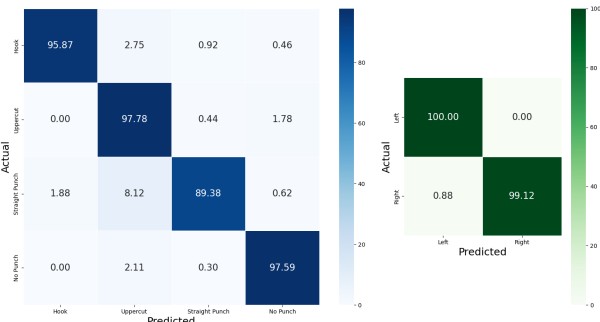

Figure 3: Confusion Matrices for Direct approach showing: punch type prediction results (left) and side classification (right).

## 6.2 STAGED APPROACH

In the staged approach, we used the DEKR-HRNet pose estimation model via the MMPose framework (OpenMMLab, 2020) to extract 17 human pose keypoints from individual video frames. As the classification model's accuracy in the staged approach is dependent on the quality of keypoint detection during the pose estimation stage, we assessed the predicted pose points against the ground truth annotation included in the dataset with the results presented in Tab. 4.

Table 4: Pose Estimation Deviations (X, Y coordinates).

| Body Part | X Deviation | Y Deviation |
|---|---|---|
| Nose | 0.714 | 5.998 |
| Left Eye | 0.742 | 0.760 |
| Right Eye | 0.627 | 1.206 |
| Left Ear | 9.186 | 6.035 |
| Right Ear | 8.345 | 4.344 |
| Left Shoulder | 22.982 | 9.164 |
| Right Shoulder | 2.583 | 21.083 |
| Left Elbow | 3.051 | 17.981 |
| Right Elbow | 5.527 | 8.992 |
| Left Wrist | 22.815 | 29.491 |
| Right Wrist | 11.922 | 5.121 |
| Left Hip | 18.819 | 72.740 |
| Right Hip | 9.871 | 64.815 |
| Left Knee | 15.663 | 23.437 |
| Right Knee | 3.292 | 1.347 |
| Left Ankle | 1.993 | 15.383 |
| Right Ankle | 2.764 | 5.113 |

As observed from pose estimation deviations, notable errors occurred in key areas such as shoulders, wrists and hips regions that are crucial for accurately identifying punch types. These deviations could be attributed to two factors: firstly, there could be some discrepancies in the definition of keypoints, such as in the case of hips; secondly, as the dataset contains fast movements during punch actions, motion blur contributes to keypoint position uncertainty, which can lead to inaccurate predicted keypoint locations. Moreover, poses adopted by actors during shadowboxing activity may be out of distribution for mainstream zero-shot models. This observation highlights the difficulty of using intermediate keypoints for action classification, meaning that challenging data may require pose model fine-tuning based on the provided annotation.

Inaccuracies in pose estimation tend to propagate throughout the pipeline, ultimately affecting final action spotting accuracy. However, these errors could potentially be mitigated by training the pose estimation model on specific training data rather than relying on zero-shot predictions, which would likely lead to more accurate keypoint detection and improved classification performance. However, optimisation of the pose estimation model is beyond the scope of this work. The punch type classifi-

cation in the staged approach achieved an accuracy of 90.50% (Tab. 5), while the side classification achieved an accuracy of 92.40% (Tab. 6).

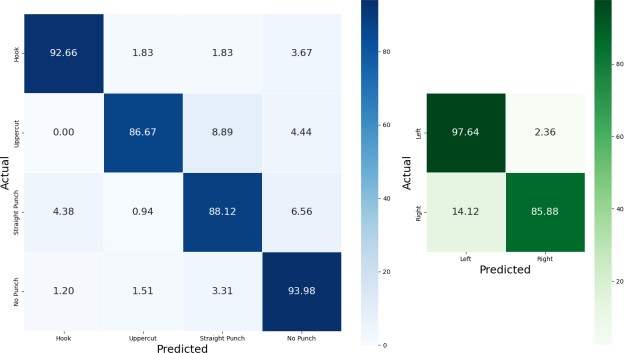

Figure 4: Confusion Matrices for the Staged approach showing: punch type prediction results (left) and side classification (right).

Table 5: Side Classification Metrics for the staged approach.

| Class | $F_1$-Score | Precision | Recall |
|---|---|---|---|
| Hook | 0.9224 | 0.9182 | 0.9266 |
| Uppercut | 0.9028 | 0.9420 | 0.8667 |
| Straight Punch | 0.8854 | 0.8896 | 0.8812 |
| No Punch | 0.9136 | 0.8889 | 0.9398 |

Table 6: Side Classification Metrics for Staged Approach.

| Class | $F_1$-Score | Precision | Recall |
|---|---|---|---|
| Left | 0.9344 | 0.8959 | 0.9764 |
| Right | 0.9097 | 0.9669 | 0.8588 |

A detailed analysis of temporal action spotting displacement is provided in Appendix A. Briefly, the temporal prediction quality follows the patterns observed in the classification metrics above. Specifically, 85.6% of correctly predicted events exhibit no temporal error (the predicted frame number matches the one annotated as correct in the ground truth data) compared to 74.3% for the two-stage approach.

## 6.3 RESULTS DISCUSSION

When comparing the direct and staged approaches, we observe that the direct method outperforms the staged approach in both punch type and side classification, achieving higher overall accuracy (94% vs. 91% for punch type classification and 99% vs. 94% for side classification). This could be attributed to the fact that the direct approach allows the model to learn directly from the raw video frames without relying on imperfect intermediate representations (human pose keypoints in this case).

Interestingly enough, side classification is less accurate for the staged approach, despite the task seeming straightforward. One possible explanation is that in some cases during a punch action, from the viewpoint prevalent in the dataset, the left hand wrist may appear closer to the right side of the video frame than the right hand wrist, and vice versa. This could confuse the temporal model used in the second step of the process.

Despite its lower accuracy demonstrated in our baseline, the staged approach offers practical advantages. By leveraging pose estimation, it provides a modular pipeline that enables not only punch recognition but also detailed analysis of biomechanics and shadowboxing technique. The quality of pose estimation plays a critical role in subsequent classification performance. Errors in keypoint

detection may directly influence the classification results. Nonetheless, the staged approach offers valuable insights into the biomechanics of punches, potentially allowing for technique improvement suggestions in computer-vision-based virtual trainer applications and more granular analysis of movement. It is worth noting that fine-tuning of the pose estimation model on the dataset annotation will likely improve the accuracy of the staged method.

In practice, the staged approach can also be advantageous in other scenarios where detailed pose analysis is essential, such as injury prevention methods, where understanding the specifics of body movement is just as important as the classification accuracy of the punch itself (Stenum et al., 2021).

### 6.4 LIMITATIONS AND FUTURE WORK

The dataset features only front-facing videos of shadowboxing, which is sufficient for punch type and side recognition, as well as for detecting some types of boxing mistakes. However, the data cannot be viewed as a general dataset for shadowboxing processing since other angles of view are not included; thus, some mistakes and intricate movements may be challenging to recognise. This makes it an open area for the next version of the dataset to include such videos.

The work also raises the question of the optimal form of data representation: using imagery data directly for end-to-end event spotting pipelines or using a staged approach with intermediate keypoints representations. Currently, we do not offer a ready answer and leave further research to the community. It is likely that the answer will depend on particular details of one's application priorities, such as flexibility for auxiliary tasks addition, inference latency, and edge device performance.

Additionally, future work could explore the impact of fine-tuning the pose-estimation model specifically on this dataset. This could provide insights into how much accuracy can be improved through tailored optimisation. Another promising avenue is exploring temporal pose prediction to better capture the dynamic nature of boxing movements over time. Furthermore, investigating the use of 3D pose predictions instead of the currently included 2D pose results may offer more comprehensive and accurate analysis of movement in a three-dimensional space.

## 7 CONCLUSION

In this work, we introduced ShadowPunch, a novel dataset designed to support advanced pose estimation, action classification, and spotting tasks within the context of boxing. The dataset comprises high-resolution videos with precise frame-level annotations for various punch types, providing a valuable resource for research into fast-paced action recognition.

We evaluated two key approaches: a direct method where models classify punch types and sides directly from video frames, and a staged approach that leverages pose estimation as an intermediate step. Our results indicate that the direct method outperformed the staged approach in terms of overall accuracy. However, the staged approach offers critical insights into body mechanics and technique, making it particularly useful for detailed sports analysis.

Our findings also highlighted the importance of reducing errors in pose estimation, as deviations in pose keypoints can significantly affect subsequent classifications. Future work could explore integrating more sophisticated pose estimation models and action recognition techniques, potentially leveraging temporal pose prediction and 3D pose predictions to enhance accuracy and comprehensiveness.

The dataset and benchmarks set a foundation for further exploration in both sports science and video-based action spotting. By addressing the limitations identified in this study, researchers can develop more robust and versatile systems both sports science and video-based fast-paced action spotting.

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

# A    APPENDIX A: TEMPORAL DISPLACEMENT RESULTS

In our method of calculating frame displacement, we infer the punch type at the target frame and compare it against the ground truth label. If the predicted punch type matches the target punch type, we proceed to compute displacement for that event. Displacement is calculated by examining a sequence of seven frames before and seven frames after the target frame (the target frame is labelled as frame 0). If the predicted punch type in any of these surrounding frames matches the target punch type and has a significantly higher probability than at frame 0 (i.e., greater than a predefined threshold ($\theta = 0.15$), the displacement is recorded based on that frame.

The displacement formula can be expressed as:

$$\Delta t = \arg \max_{t \in [-7,7]} \begin{cases} P(punch\_type_t) \\ \text{such that } punch\_type_t = punch\_type_0 \\ \text{and } P(punch\_type_t) > P(punch\_type_0) + \theta \end{cases}$$

where: - $\Delta t$ is the frame displacement, - $P(punch\_type_t)$ is the probability of the punch type at frame $t$, - $punch\_type_t$ is the predicted punch type at frame $t$, - $punch\_type_0$ is the predicted punch type at the target frame, - $t = 0$ represents the target frame, and - $\theta$ is the threshold (in this case, 0.15).

This evaluation method can be refined further depending on the model architecture. Notably, displacement at larger frames (e.g., ($t = 7$)) could be attributed to various phenomena such as unfiltered cases with two punches with a very small gap in time or fast movements. In such cases, more complex heuristics can be developed to capture and handle these events.

## A.1    ONE-STAGE APPROACH DISPLACEMENT RESULTS

Table 7: One-Stage Approach Displacement Results

| Displacement (Frames) | Count | Percentage (%) |
|---|---|---|
| 0 | 840 | 85.6 |
| 1 | 25 | 2.6 |
| 2 | 11 | 1.1 |
| 3 | 14 | 1.4 |
| 4 | 20 | 2.0 |
| 5 | 13 | 1.3 |
| 6 | 12 | 1.2 |
| 7 | 23 | 2.3 |

## A.2    STAGED APPROACH DISPLACEMENT RESULTS

Table 8: Staged Approach Displacement Results

| Displacement (Frames) | Count | Percentage (%) |
|---|---|---|
| 0 | 694 | 74.3 |
| 1 | 12 | 1.3 |
| 2 | 12 | 1.3 |
| 3 | 16 | 1.7 |
| 4 | 14 | 1.5 |
| 5 | 16 | 1.7 |
| 6 | 23 | 2.5 |
| 7 | 46 | 4.9 |

# B    APPENDIX B: DATASET DATA SAMPLES

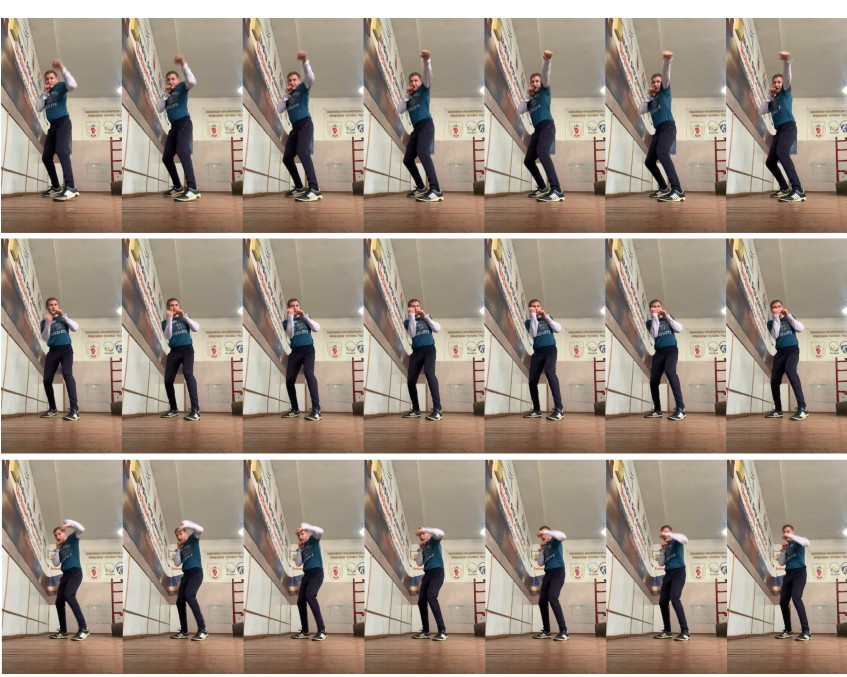

Figure 5: Example of three punch types frame sequences from the ShadowPunch dataset: Straight – top, Uppercut – middle, and Hook – bottom.

