# OpenReview forum: "ShadowPunch: fast actions spotting benchmark"
_ICLR.cc/2025/Conference — ICLR 2025 Conference Withdrawn Submission_

### Official Review · Reviewer_y3sE · 2024-10-29

**Soundness:** 2
**Presentation:** 3
**Contribution:** 2
**Rating:** 5
**Confidence:** 4

**Summary:**

This paper presents a shadowboxing video dataset comprising 27 videos with a total of 230,502 frames. It includes 4,840 annotated, fast-paced boxing events and 4,761 frames, each labeled with 17 keypoints. To facilitate further research, the authors provide several baseline neural networks, covering both direct prediction and two-stage approaches.

**Strengths:**

(1) The dataset provides 17 keypoints for each of the 4,761 frames, delivering detailed data suitable for training and evaluating pose estimation and action recognition models.

(2) By specifically targeting fast-paced boxing events, the dataset addresses a challenging area for event spotting.

(3) The authors supply baseline neural networks, including both direct prediction and two-stage approaches, which allow for easy benchmarking and comparison.

**Weaknesses:**

(1) The dataset includes only four action classes—hook, undercut, straight, and no punch—making it a relatively simple task, as shown by the high F1 scores of baseline models. This limited action set restricts potential for further improvement. Expanding the dataset to include more actions involving body and foot movements, such as ducking, slipping, and rolling punches, would enhance its complexity and usefulness.

(2) The dataset appears to contain data captured only from a front-facing camera, which limits its challenge and versatility. Including side or rear camera views would make the dataset more challenging and valuable for developing models that can generalize across different perspectives.

(3) Details on the keypoint annotation process and quality control are lacking. Information on how keypoint annotations were verified or consistency-checked would increase confidence in the dataset's accuracy and reliability. For example, details about the annotation tool used, the number of annotators involved, and any measures of inter-annotator agreement would be valuable.

**Questions:**

(1) How was the quality of keypoint annotation ensured in the dataset? Could you describe the processes used for verifying and validating the accuracy of these annotations?

(2) Were sensors used in locating the keypoints? Are the 3D coordinates of the keypoints available?

---

### Official Review · Reviewer_PFwL · 2024-11-03

**Soundness:** 2
**Presentation:** 2
**Contribution:** 1
**Rating:** 3
**Confidence:** 5

**Summary:**

The authors provide a dataset for fighting (Shadow punching) and some preliminary results using standard algorithms

**Strengths:**

An annotated dataset for shadow punching (boxing, fighting) is provided.

**Weaknesses:**

It is not clear how this data was annotated, who did the annotations, were they experts?
The paper would have been interesting if there was an automated way to do the annotations on this dataset.
This dataset is not a contribution.
It appears that the scenes are staged in a gym.

**Questions:**

How many fighters were used, were the video sequences taken from different fights, gyms?
It appears there is only 1 actor, in a fight there are 2 which complicates things and makes for more interesting and relevant training data.
Who did the annotations, were they experts?
Please define what you mean by event and action, clear definitions are required?  IS an event an instance in time and an action takes some time? unclear
Does Table 1 define all the events you are looking at.
There are many datasets for pose estimation already, why would we use this?
How accurate are the annotations?
Again, need to define what the actions are?
What is a spotting task in boxing, unclear for those of us that do not follow boxing?

---

### Official Review · Reviewer_C5Lk · 2024-11-04

**Soundness:** 1
**Presentation:** 1
**Contribution:** 1
**Rating:** 3
**Confidence:** 4

**Summary:**

Thanks the authors for the efforts on making this submission to ICLR2025.

Summary:
This paper claims a contribution to video understanding research with a novel dataset specific for **rapid** video action detection (i.e., action events happening in short period of time). The dataset is collected from shadow-boxing - a type of popular sports, and contains frame-level annotations of action types and pose keypoints. To facilitate future research, this paper provides benchmark results on action detection using two methods - a simple baseline and a more complicated approach using pose estimation as auxiliary task.

**Strengths:**

- It is really hard to find strengths for this submission.

**Weaknesses:**

Weakness:
- (**critical**) Please be crystal clear on the defintion of fast action - what is considered as fast for action detection; how does comtemporay work perform on fast action detection.

- (**critical**) below average writing quality: Typos & Mistaken in writing / presentation as follows.
    - Line 315, [0..1] -> [0, 1],
    - Table 1 exceeds the right boundary of regular writing space.
    - Equations should end with period.


- (minor) The paragraphs on **Public datasets** and **Video understanding models** in the related work section are overly detailed and stray from the main topic of this submission. The writing for F1 score and Accuacy evaluation metrics is also excessively unnecessary.


- (minor) It's difficult to grasp why a well-annotated shadow-boxing dataset is considered urgently necessary for video understanding research, as mentioned in lines 146-147. A study can be done to shed more light might be: Showcase the video understanding models trained on the proposed shadow-boxing dataset can lead to performance improvements on other video understanding tasks / datasets.


- (minor) On **comprehensive boxing dataset** in line 57, the following critical stats of the dataset is missing.
    - The ratio of gender.
    - The number of different performer.
    - The number of different indoor background and outdoor background.
    - The average frame-span per **fast** boxing actions. Line 208 vaguely states ` more than one event per second`. Yet, the fps of this dataset is **60**, so the number of frames per action can still be quite a lot. It is worth clarifying.


- (minor) On `the model can learn a comprehensive representation of temporal dynamic` in line 252, it is either overstated or missing some context. It is quite impossible to lean a comprehensive temporal dynamic representation based on the shaow-boxing dataset alone, given there exist so many types of temporal dynamics, e.g., daily life actions, professional sport actions, etc.


- (**critical**) Given the results of Table 2 & 3, it is quite unobvious what is left to be researched with this work - the classification results are near perfection.

-  (**critical**) No interpretation study has been conducted, e.g., shuffle the frames to ensure that the model learns temporal dynamics.

**Questions:**

- The choice of using `ResNet34-3D` as the baseline architecture brings concerns. Why not more advanced models?

---

### Official Review · Reviewer_ht8R · 2024-11-04

**Soundness:** 3
**Presentation:** 3
**Contribution:** 1
**Rating:** 1
**Confidence:** 4

**Summary:**

The paper proposes a dataset of shadow boxing activities, where each video shows an individual pretending to punch someone (who is not in the video). There are three types of punching, recorded in 27 videos annotated by boxing experts. Baseline experiments are performed on the videos using reasonable, current methods and achieve more than 99% accuracy.

**Strengths:**

The videos are continuous, including many instances of the punch types. They are annotated by experts in order to properly differentiate the three fine-grained classes.

The paper is clearly written, with a reasonable description of the dataset.

Baseline performance bounds are measured using a method that inputs pixels directly, and another that inputs pose information. As in other domains, pixel inputs are more accurate.

**Weaknesses:**

The topic of the paper, shadow boxing video event detection, is very narrow and unlikely to be of significant interest to the ICLR community. There are many dozens of video activity detection datasets in the literature. This dataset contains a very small number of activities by current standards (3 proposed here vs. dozens or hundreds in other datasets), collected from similar conditions that make the problem very easy. There is a single individual in the scenes, there are no other activities happening, and the subject is large in the videos. The scene variety is reasonable, with different settings such as people's homes vs. boxing gyms, but this is not sufficient to overcome the limitations of the dataset.

Performance on the dataset is very accurate, greater than 99% with existing baselines. Datasets that are already solved are generally not useful to the research community since there is no research required to improve performance. Typically new datasets should have accuracies below 50%.

There are ethical concerns because the dataset includes human subjects but there is no mention of HSR compliance or IRB review. If this dataset is released to the public, as stated in the paper, then researchers must have assurance that the dataset was collected with the proper ethical safeguards.

Because of these limitations and concerns, this paper is likely to be of little interest to the ICLR audience.

**Questions:**

Was the dataset reviewed by an IRB? Was consent obtained from the subjects shown in the video?

**Details Of Ethics Concerns:**

The paper proposes a dataset for human activity recognition, including videos showing facially recognizable subjects. The paper does not mention anything about obtaining permission from the subjects, review by an IRB, or any acknowledgement of typical issues with datasets showing human subjects.

---

> ### Comment · Reviewer_ht8R · 2024-11-26
>
> No rebuttal was submitted. My score remains unchanged.

---

### Note · Authors · 2024-11-28

**Comment:**

Thank you for taking the time to read and provide reviews on our work, we appreciate it.

**Withdrawal Confirmation:**

I have read and agree with the venue's withdrawal policy on behalf of myself and my co-authors.